# Optimization of Hyperglycemic Induction in Zebrafish and Evaluation of Its Blood Glucose Level and Metabolite Fingerprint Treated with *Psychotria malayana* Jack Leaf Extract

**DOI:** 10.3390/molecules24081506

**Published:** 2019-04-17

**Authors:** Khaled Benchoula, Alfi Khatib, Fairuz M. C. Quzwain, Che Anuar Che Mohamad, Wan Mohd Azizi Wan Sulaiman, Ridhwan Abdul Wahab, Qamar Uddin Ahmed, Majid Abdul Ghaffar, Mohd Zuwairi Saiman, Mohamed F. Alajmi, Hesham El-Seedi

**Affiliations:** 1Department of Basic Medical Sciences, Kulliyyah of Pharmacy, International Islamic University Malaysia, Kuantan 25200, Pahang, Malaysia; benchoulakhaled@hotmail.fr (K.B.); dranuar@iium.edu.my (C.A.C.M.); drwanazizi@iium.edu.my (W.M.A.W.S.); 2Department of Pharmaceutical Chemistry, Kulliyyah of Pharmacy, International Islamic University Malaysia, Kuantan 25200, Pahang, Malaysia; quahmed@iium.edu.my (Q.U.A.); jd.majid@yahoo.com (M.A.G.); 3Faculty of Medicine, Universitas Jambi, Jambi 36122, Indonesia; 4Department of Biomedical Science, Kulliyyah of Allied Health Science, International Islamic University Malaysia, Kuantan 25200, Pahang, Malaysia; ridhwan@iium.edu.my; 5Institute of Biological Sciences, Faculty of Science, University of Malaya, Kuala Lumpur 50603, Malaysia; zuwairi@um.edu.my; 6Department of Pharmacognosy, College of Pharmacy, King Saud University, Riyadh 11451, Saudi Arabia; malajmii@KSU.EDU.SA; 7Division of Pharmacognosy, Department of Medicinal Chemistry, Biomedical Centre, Uppsala University, Box 574, SE-751 23 Uppsala, Sweden

**Keywords:** Type 1 diabetes, zebrafish, *Psychotria malayana*, alloxan, insulin, streptozotocin

## Abstract

A standard protocol to develop type 1 diabetes in zebrafish is still uncertain due to unpredictable factors. In this study, an optimized protocol was developed and used to evaluate the anti-diabetic activity of *Psychotria malayana* leaf. The aims of this study were to develop a type 1 diabetic adult zebrafish model and to evaluate the anti-diabetic activity of the plant extract on the developed model. The ability of streptozotocin and alloxan at a different dose to elevate the blood glucose levels in zebrafish was evaluated. While the anti-diabetic activity of *P. malayana* aqueous extract was evaluated through analysis of blood glucose and LC-MS analysis fingerprinting. The results indicated that a single intraperitoneal injection of 300 mg/kg alloxan was the optimal dose to elevate the fasting blood glucose in zebrafish. Furthermore, the plant extract at 1, 2, and 3 g/kg significantly reduced blood glucose levels in the diabetic zebrafish. In addition, LC-MS-based fingerprinting indicated that 3 g/kg plant extract more effective than other doses. Phytosterols, sugar alcohols, sugar acid, free fatty acids, cyclitols, phenolics, and alkaloid were detected in the extract using GC-MS. In conclusion, *P. malayana* leaf aqueous extract showed anti-diabetic activity on the developed type 1 diabetic zebrafish model.

## 1. Introduction

Zebrafish have been used effectively for drug screening [1]. The advantages of utilizing zebrafish as an animal model are as follows: lower maintenance costs, shorter periods of testing, easily controlled experimental conditions, and most importantly, the genetic similarity between zebrafish and humans (approximately 70%) [2]. The small size of zebrafish (3–5 cm) allows them to be kept in small tanks [3]. Zebrafish as a vertebrate model shows significant similarity to human physiology. The same carbohydrate-regulated genes in mammals have been detected in zebrafish. Moreover, the zebrafish pancreas has the same functions as its mammalian counterpart regarding glucose homeostasis, including producing and secreting insulin, glucagon, somatostatin and digestive enzymes such as amylase [4]. Furthermore, more than 14,000 genes of zebrafish genome have been studied exclusively and it is fully sequenced. Consequently, it contributes to the establishment of diabetes transgenic model [5]. Despite its advantages, however, working with zebrafish requires handling skills to extract the organs for histological purposes [6].

Considering their advantages, zebrafish have been chosen as an ideal animal model for different diseases including diabetes. Nevertheless, the conditions for inducing type 1 diabetes in adult zebrafish vary depending on unpredictable factors. Thus, an optimized protocol for establishing a type 1 diabetes zebrafish should be developed prior to testing the anti-diabetic activity of potential treatments. 

Certain products have shown promising results as hyperglycaemia drugs through screening using in vitro tests. However, some of these potential drugs lost their anti-diabetic activity when tested in mammalian animal models due to pharmacokinetic issues and/or toxicity. In this stage, the zebrafish model is could be used as an alternative screening tool prior to testing on mammalian models [7]. 

*Psychotria malayana* Jack is a plant that is widespread throughout tropical and subtropical countries. These plants have been used traditionally to treat gastrointestinal disease, stomach-ache and infections of the female reproductive system in countries such as India, Indonesia, and Brazil. Several studies on the antioxidant, anti-inflammatory and antimicrobial properties of this plant have been reported [7,8,9]. Due to its curative properties, the species of *Psychotria* herb are used traditionally for treating diabetes. However, the scientific proof of this plant on diabetes treatment is still lacking. Although the phytochemical studies of *Psychotria* species showed the presence of alkaloid compounds in the aerial parts of the plant including calycanthine, chimonantine, hodgkinsine, and *N*-methyltryptamine [7], none of these compounds have been reported to exhibit anti-diabetes property. The available approaches for treating diabetes include dietary control, insulin injection, and oral diabetes medications. However, oral medications have been confirmed to be inadequate and causing several side effects, while insulin injection solves the problem only temporarily [10]. Moreover, hypertrophy as a result of many insulin injections in a body area is another complication of insulin therapy [11]. This condition leads to the necessitates finding new drugs or treatments with fewer side effects. 

Thus, the aim of this study was to develop a type 1 diabetic adult zebrafish model through chemical induction and to evaluate the anti-diabetic activity of *P. malayana* leaf extract on the developed model.

## 2. Results

### 2.1. Development of the Induced Type 1 Diabetic Zebrafish Model

#### 2.1.1. Induction of Type 1 Diabetes Using Streptozotocin

Intine et al. [12] reported that the established protocol using streptozotocin (STZ) as an inducer (STZ multiple injections at different time points) with a dose of 300 mg/kg caused an increment of blood glucose level in the zebrafish with low mortality (5%). However, our study showed high mortality (80%) and hypoglycaemia for all samples using this method, in which the level of blood glucose significantly dropped (*p* < 0.05) to 21.5 ± 2.1, 22.3 ± 3.3, and 23.8 ± 4.2 mg/dL at 24, 48, and 72 h after STZ injection, respectively (Figure 1). There were no significant differences among the glucose levels at different time points (*p* > 0.05) but the glucose levels were significantly different between the STZ treated and the healthy group (*p* < 0.05).

This unexpected result led us to perform an experiment on the establishment of the type 1 diabetes zebrafish model prior to testing the anti-diabetes activity of the plant extract. Different doses of STZ (100, 300, 500, and 700 mg/kg) were tested as shown in Figure 2. No mortality was observed in any of the zebrafish groups. Furthermore, there was no significant difference between the healthy fish and the zebrafish induced with 100 mg/kg STZ. In contrast, the STZ injected groups at the higher doses were significantly different from the healthy group (*p* < 0.05). In addition, the blood glucose levels among the 3 groups injected with higher doses of STZ were not significantly different. Hence, 300 mg/kg STZ was selected for further optimization work because this dose was the lowest among the effective doses. 

Further optimization experiment aimed to evaluate the effect of the selected doses on the fasting blood glucose level of zebrafish at different timelines (24, 48, and 72 h) after STZ injection. As shown in Figure 3, 300 mg/kg STZ was able to significantly (*p* < 0.05) increase the fasting blood glucose level of the zebrafish for up to 24 h compared to that of the healthy group. Moreover, this STZ dose did not cause mortality on the zebrafish. Nevertheless, this result was unsatisfactory because the blood glucose level dropped rapidly. Hence, a further optimization experiment to prolong the period of hyperglycaemia was performed by exploring another induction agent, alloxan.

#### 2.1.2. Induction of Type 1 Diabetes Using Alloxan

Various doses of alloxan (100, 200, 250, 300, and 350 mg/kg) were tested on the effect on the fasting blood glucose level. Figure 4 shows that 250 and 300 mg/kg alloxan were able to significantly (*p* < 0.05) elevate the blood glucose level at 24 h after the alloxan injection. Furthermore, at 48 h after the injection, the blood glucose level of the zebrafish injected with 300 mg/kg alloxan remained elevated and significantly different (*p* < 0.05) from that of the healthy fish. The mortality rate at the 100, 200, and 250 mg/kg doses were 0%, and although the mortality rate at the 300 mg/kg dose was 10%, it was still acceptable [13]. However, the mortality rate at the 350 mg/kg dose was too high (70%). Hence, the permitted dose should not exceed 300 mg/kg. Based on this result, 300 mg/kg was used for evaluation of the hyperglycaemia inducing effect of alloxan over a longer experimental period (8 days).

Figure 5 shows that blood glucose levels were significantly higher (*p* < 0.05) in most of the alloxan treated zebrafish than in the healthy fish between 24 h (1 day) and 168 h (7 days) after injection with 300 mg alloxan/kg. The mortality rate was 10% which was still acceptable. 

### 2.2. Effect of Glibenclamide and Insulin as a Positive Control on Fasting Blood Glucose Level of the Diabetic Zebrafish Model

Glibenclamide and insulin were used as positive controls for the developed method. Both drugs were injected to evaluate their capability to reduce the fasting blood glucose level of the diabetic zebrafish. As shown in Figure 6, a glibenclamide dose of 1.5 mg/kg was not effective in reducing the fasting blood glucose level of the diabetic zebrafish, since the blood glucose level was still significantly higher (*p* < 0.05) than that of the healthy group. Nevertheless, no mortality was observed for the glibenclamide treated zebrafish, while the mortality rate of the diabetic group was 10%.

Figure 7 indicates that an insulin dose of 1 U/kg was effective in reducing the blood glucose to normal levels after at 24 h after the alloxan injection, since no significant difference (*p* < 0.05) was observed in glucose levels between the diabetic group and the healthy group. The mortality rate of the treated zebrafish was 0% indicating an improvement compared to the diabetic group with a mortality rate of 10% due to alloxan injection.

### 2.3. Evaluation of the Anti-Diabetic Activity of Psychotria Malayana Leaf on the Alloxan Induced Type 1 Diabetes Zebrafish

Different doses of the aqueous extract of *P. malayana* leaf (1, 2, and 3 g/kg) were tested on the alloxan-induced type 1 diabetic zebrafish model. Figure 8 indicates that all doses were effective in lowering blood glucose levels which were significantly decreased compared to those in diabetic zebrafish at 48 h after alloxan injection and 24 h after the feeding with the plant extract. Plant extract at 1 and 3 g/kg was able to reduce the blood glucose level to normal levels. Nevertheless, the blood glucose level of the zebrafish treated with the dose of 2 g/kg was still significantly different from that of the healthy fish (*p* < 0.05). The positive control, insulin injection, was able to lower the blood glucose level to normal levels. No mortality was observed in the healthy and treated groups. The mortality rate of the diabetic group was 10%.

### 2.4. GC-MS Analysis of the *P. malayana* Leaf Extracts

Table 1 shows 29 compounds identified in the *P. malayana* leaf aqueous extract analyzed by GC-MS. All fragmented MS spectra of each ion were compared to the NIST14 database library. Thus, only those compounds with similarity indices higher than 90% are listed in Table 1. Sugars such as sucrose, d-mannose, d-fructose, d-galactose, d-allose, d-trehalose, d-psicose, d-tagatose, and l-sorbose are among the major compounds present in the extract and dominate more than 50% of the total peak area. The proportion of minor compounds is shared by different groups of compounds such as phytosterols (beta-sitosterol), sugar alcohols (myo-inositol, xylitol, ribitol, and erythritol), sugar acid (xylonic acid), free fatty acids (stearic acid and palmitic acid), cyclitols (quinic acid and shikimic acid), phenolics (benzenetriol, α-tocopherol, and dihydroxyphenylglycol), and alkaloid (piperidine).

### 2.5. LC-MS Based Fingerprinting of Zebrafish Serum

The collected serum from different groups of zebrafish (healthy group, diabetes group, insulin-treated group, and the group treated with the plant extract at doses of 1, 2, and 3 g/kg were analyzed using LC-MS to investigate their serum fingerprints. Multivariate data analysis with partial least square-discriminant analysis (PLS-DA) was performed on the pre-processed LC-MS dataset to observe the discrimination among the samples. The best fit was achieved with *R*^2^*Y*_(cum)_ = 0.456 and *Q*^2^_(cum)_ = 0.243, where 45.6% of the variation was explained and model predictability was acceptable [14]. Validation of the multivariate calibration was performed using permutation testing. According to [14] the model is valid if the intercepts of R^2^ and Q^2^ are less than 0.4 and −0.05, respectively. The R^2^ and Q^2^ values in this study were 0.225 and −0.243, respectively, which were in an acceptable range. as shown in Figure 9.

Figure 10 shows the score scatter plot displaying the discrimination of the healthy group (H), diabetes group (D), insulin-treated group (I), and groups treated with the plant extract at doses of 1 g/kg (1), 2 g/kg (2) and 3 g/kg (3). The diabetic group was separated from the healthy group alongside Partial Least Square-Discriminant Analysis (PLS-DA) component 2. Furthermore, the group treated with the plant extracts and insulin were shifted closer to the healthy group by PLS-component 2. Interestingly, PLS-DA component 1 separated the plant extract treated group from the healthy, diabetic, and insulin-treated groups. The higher dose of the plant extract shifted the sample closer to the healthier state.

### 2.6. Histological Examination

Analysis of paraffin sections of pancreatic islets was performed using haematoxylin and eosin staining of sections obtained from the healthy, diabetic, and insulin-treated zebrafish and the fish treated with 3 g/kg plant extract. The islets cells in the healthy group showed a normal structure of (Figure 11a,b). However, both diabetes (Figure 11c,d) and treated zebrafish (Figure 12) showed different islet structure compared to the healthy zebrafish, and there was a considerable loss of cells inside the islets.

Analysis of paraffin-embedded sections of zebrafish liver was performed using haematoxylin and eosin staining of sections obtained from the healthy, diabetic, and insulin-treated zebrafish and the zebrafish treated with 3 g/kg plant extract. All hepatocyte structures in all zebrafish were normal except for those in the insulin-treated zebrafish (Figure 13, Figure 14, Figure 15 and Figure 16), which appeared as nodules of damaged hepatocytes separated by fibrous bands (Figure 15). 

## 3. Discussion

### 3.1. Development of the Induced Type 1 Diabetes Zebrafish Model

The established protocol for inducing type 1 diabetes in zebrafish reported by Reference [12] failed to elevate the fasting blood glucose of the zebrafish in this study. This method caused hypoglycaemic conditions and high mortality, in contrast to the expected results as reported. The high mortality could be due to: (1) multiple injections irritated the cavity of the fish and induced skin inflammation; (2) the long duration of the experiment caused the fish experience stress; (3) the large volume of STZ injected over almost 20 days might have induced toxicity in the fish organs such as the liver [15]. In addition, the injected fish reduced their food intake which is a positive correlation to the drop of the blood glucose levels [16].

These unexpected results led us to optimize the induction condition for the type 1 diabetes zebrafish model. In this study, the intraperitoneal-administration of STZ and alloxan were able to elevate the blood glucose level. The effective period of the alloxan induction at the optimized dose (300 mg/kg) was 7 days. The blood glucose level in the induced fish was significantly (*p* < 0.05) higher than the normal fasting blood glucose level in zebrafish, which is 50–75 mg/dL according to Reference [17]. The effective period of alloxan induction was longer than that of STZ induction at a higher dose (500 mg/kg) which lasted only 4 days, although STZ induction did not cause mortality. The mortality rate of the zebrafish induced by 300 mg/kg alloxan was 10%, which was still acceptable [13]. Increasing the alloxan dose (to 350 mg/kg) caused an unacceptable mortality rate (70%). 

Similar work on the use of alloxan has been reported by Reference [18]. However, the detailed information on the duration of the effective period was uncertain. In addition, data on the histological examination of the affected pancreas and liver were not provided, leading to difficulty in determining whether the model for type 1 diabetes had been established. Moreover, the exposure method was not accurate in terms of dose, since the fish were immersed in the alloxan solution. Furthermore, Moss et al. (2009) reported the impact of high dose STZ on the capability of zebrafish to regenerate the pancreas tissue, however, its effect on the blood glucose level was not a major concern [19]. 

The diabetogenic characteristic of alloxan was reported by Reference [20]. Alloxan has a chemical structure similar to that of glucose, which helps in penetrating the lipid bilayer of the β-cells of the pancreas by binding to the GLUT2 glucose transporter [21]. Alloxan selectively destroys β-cells in two ways. The reduction of alloxan in the cytosol of β-cells can produce oxygen species, which play a significant role in the necrotic process of β-cells. The reduction can take place when alloxan reducing agents contain SH groups, such as glutathione, and cysteine, and in certain cases when ascorbate is present in the cytosol. Furthermore, the inhibition of glucokinase has been considered the secondary effect of alloxan in the β-cells [22]. Glucokinase is the first enzyme in glycolysis and is involved in converting glucose to glucose-6-phosphate, openings the possibility for enormously complex pathways [23]. 

Both alloxan and STZ cross the phospholipid bilayer of β-cells, via glucose transporter GLUT2 [24]. However, the effect of STZ is completely different from alloxan. The main effect of STZ in β-cells is the alkylation of DNA leading to DNA fragmentation [15]. The fragmentation of DNA provokes the repair enzyme, poly (ADP-ribose) polymerase to repair DNA damage. The over-activation of this enzyme can decrease the ATP stores of the cells causing β-cell necrosis. In addition, STZ can be a nitric oxide (NO) donor. This feature increases the level of cGMP, as a result of the increase in the activity of guanylyl cyclase [25]. The short effect of both alloxan and STZ is due to the high regeneration of the β-cells in zebrafish. Zebrafish have the capacity to transdifferentiate the α-cells to β-cells in the ablation case of β-cells ablations through inhibition of insulin-like growth factor (IGF) pathway by increasing secretion of insulin-like growth factor binding protein 1 (Igfbp1) from the liver of zebrafish larvae [26]. Another finding indicated that the centroacinar cells (CACs), a type of ductal cell in adult zebrafish, could contribute to the process of β-cell regeneration when β-cells were ablated [27].

### 3.2. Effect of Glibenclamide and Insulin on Fasting Blood Glucose Level in Diabetic’s Zebrafish Model

Glibenclamide is a second generation sulfonylurea drug [28]. It is an oral anti-diabetic; that is utilized to treat hyperglycemia. The mechanism of action for glibenclamide is specifically related to β-cells in the pancreas, and it upgrades insulin secretion by connecting to a particular protein on the surface and prompting the activation of the potassium channels which are ATP-sensitive. The cell membrane will be depolarized caused by an increase in potassium inside the β-cells, and hence, the calcium channels open, enabling calcium to enter the cytosol of β-cells. Notably, the increase in intracellular calcium promotes a further combination of insulin granules with the cell membrane and releases insulin [29]. In this investigation, glibenclamide was unable to reduce the blood glucose level of the diabetic zebrafish, which might be caused by an inability of the drug to induce insulin production due to a major loss of cells in the islets of Langerhans in diabetic fish (Figure 7). The drug was unable to promote repair of the damaged β-cells and therefore failed to induce insulin production through the aforementioned mechanism [30]. In contrast, insulin injection was able to lower and normalize the blood glucose level of the diabetic zebrafish as expected (Figure 8). Notably, insulin injection does not require healthy β-cells for this effect. Insulin stimulates the translocation of glucose transporter GLUT4 to the plasma membranes of liver and fat cells [31]. The result of this study is in accordance with Reference [32] which reported that insulin injection reduces fasting blood glucose levels in zebrafish with hyperglycemia induced by immersion in a glucose solution. 

### 3.3. Anti-Diabetic Activity of *P. malayana* Leaf in Alloxan-Induced Type 1 Diabetic Zebrafish

High doses of the plant extract (> 1 g/kg) and a single episode of force-feeding were applied in this study, considering that multiple feedings were impossible due to the irritation of the upper digestive tract caused by the feeding procedure. The plant extract was able to lower and normalize the blood glucose levels of alloxan-induced type 1 diabetic zebrafish in the present study. No significant differences (*p* > 0.05) were observed in the blood glucose levels between the healthy zebrafish group and the groups treated with the plant extract at doses of 1, 2, and 3 g/kg (Figure 9). This analytical approach is called a targeted or un-holistic analytical approach because only a targeted metabolite (in this case, was glucose) was analyzed. Nevertheless, this approach might overlook the overall ability of the plant extract to normalize all metabolites in the serum. Hence, LC-MS based fingerprinting with multivariate data analysis was applied to evaluate the overall effect of this plant extract on type 1 diabetic zebrafish. 

A supervised multivariate data analysis, PLS-DA, was used to process the LC-MS data set of the samples. Figure 12 shows that PLS-DA was able to detect that different doses of the plant extract on modifying the serum metabolite profile closer to that of the healthy state, which was not observed using the targeted analysis approach. At all doses, the plant extract was able to shift the serum profile of the diabetic fish closer to that of the healthy group alongside the PLS-DA component 2. PLS-DA component 1 showed different effects of different doses of the plant extract in normalizing the serum profile of the fish. The zebrafish treated with the plant extract at a dose of 3 g/kg was the closest to the healthy group based on PLS-DA component 1 compared to the other doses. In addition, the zebrafish group treated with insulin injection was the closest and very similar to the healthy group among all treatments.

There are enormous medicinal plants showed a positive effect in treating diabetes [33,34]. These plants have possible mechanisms for the plant extract to lower the blood glucose level: lowering glucose production in the liver, reducing glucose absorption in the intestine, increasing peripheral glucose uptake in the muscle and adipose tissue, or increasing insulin production in the β-cells [35]. The last mechanism was not likely in the present study since this plant extract cannot repair damaged β-cells as proven by the histological study. *Psychotria* sp. are able to inhibit α-amylase but was unable to inhibit α-glucosidase [36]. Moreover, inhibition of both enzymes was correlated with lowering blood glucose levels [37]. A similar result was reported in [38], an investigation of the effect of *Petroselinum crispum* extract on diabetic rats. *P. crispum* extract was able to normalize blood glucose levels, but the histopathological investigation showed no effect on the damaged β-cells in rat pancreas. The authors suggested that the mechanism of action of this plant involved the inhibition of gluconeogenesis and the stimulation of glycolysis.

Histological examination was performed in the present study to evaluate the effect of alloxan on the pancreas of the diabetic zebrafish compared to those of the healthy zebrafish using H&E staining of paraffin sections. Figure 11 shows that the islets of Langerhans in alloxan injected zebrafish demonstrated a considerable decrease in endocrine islet cellularity as opposed to the healthy zebrafish. 

GC-MS detected some compounds in this plant extract (Table 1) which are reported to have anti-diabetes activity. The free fatty acids stearic acid and palmitic acid [39,40] as well as the sugar alcohols myo-inositol, ribitol, and erythritol were reported to have α-glucosidase inhibitory activity [41,42]. Other detected groups of compounds such as phytosterol (beta-sitosterol), cyclitols (quinic acid and shikimic acid), 1-monopalmitin, and glycerol monostearate were also reported as α-glucosidase inhibitors [43,44,45]. Myo-inositol and quinic acid were highly abundant in this extract with peak areas of 5.94% and 1.31%, respectively. Another reported anti-diabetes compound detected in this extract is α-tocopherol. Bursell and King [46] reported that the activity of α-tocopherol decreases the levels of diacylglycerol and protein kinase C induced in diabetes or hyperglycaemic conditions. 

α-Glucosidase plays an important role in the digestion of complex carbohydrates by cleaving oligosaccharides into monosaccharides and is responsible for the final step in the digestive process of carbohydrates that eventually leads to postprandial hyperglycemia. The enzyme’s inhibitors will compete with the oligosaccharides for the binding site thus they are classified as the classic competitive inhibitors. An α-glucosidase inhibitor is a preferred agent in the management of postprandial hyperglycemia in type-2 diabetes mellitus [47]. Detection of various α-glucose inhibitors in the *P. malayana* extract could explain one possible mechanism by which the extract to lower blood glucose levels via inhibition of carbohydrate digestion and not through the recovery of β-cells in the pancreas, as proven by histological examination.

## 4. Materials and Methods

### 4.1. Chemicals

Streptozocin and alloxan were purchased from Sigma Aldrich^®^ (Taufkirchen, Germany). Glibenclamide (5 mg for each tablet) was obtained from Aventis Phaema^®^ (Jakarta, Indonesia). Whilst, xylene, haematoxylin, eosin, Disteryne, plasticizer and xylene mixture and all organic solvents were purchased from Fisher Scientific^®^ (New Hampshire, UK). Finally, formaldehyde was procured from Merck KGaA^®^ (Frankfurt, Germany). Human insulin from Novolin^®^ (Bagsvaerd, Denmark)

### 4.2. Maintenance of the Fish

Mixed-sex adult zebrafish (*Danio rerio*) at 3-months of age were purchased from a local supplier (Three B Aquatics Sdn. Bhd, Kuala Lumpur, Malaysia) and maintained in 9-L acrylic tanks (50 fish per tank) in a closed, multirack aquatic housing system at Kulliyyah of Pharmacy, International Islamic University Malaysia. All tanks were supplied with well-aerated dechloraminated reverse osmosis (RO) water at 28 (± 0.1) °C. The water was maintained at pH 6.8–7.5 and salinity 800–1200 µS/cm and circulated through different filters (120-µm filter pad, 50-µm canister filter, biological filter, active carbon absorption filter and UV disinfection filter) before reaching the tanks. Fish were maintained under a light/dark cycle of 14 h/10 h and fed twice daily using an adult zebrafish complete diet (Zeigler Bros Inc, Gardners, PA) containing 55% crude protein, 15% crude fat, 1.5% crude fiber and 12% water. 

### 4.3. Induction of Type 1 Diabetes Using Streptozotocin

A protocol described by Intine et al. [12] was followed to induce diabetes in zebrafish. The fish were anesthetized by exposure to water at 15 and 5 °C for a few seconds prior to streptozotocin (STZ) injection. STZ solution was prepared by dissolving 10 mg of STZ in 1 mL of citrate buffer (pH 4.7). The fish were injected intraperitoneally with 300 mg/kg STZ solution on days 1, 3, 5, 12, and 19. Unfortunately, the fish developed hypoglycaemia, and mortality was high; thus, the following modified induction protocol was used to improve the result. The fish were fasted for 12 h (overnight) prior to STZ injection. The anaesthetized fish were injected intraperitoneally with STZ at 4 different doses (100, 300, 500 and 700 mg/kg). The healthy control group was injected with citrate buffer (pH 4.7). Subsequently, after the injection, the fish were fasted for 24 h prior to blood collection. The blood was withdrawn from the heart using a syringe; the needle was pushed carefully under the gill, and approximately 10 µL of blood withdrawn. Lastly, the blood glucose level was measured using a glucometer (Easy Touch^®^ GCU glucometer, Miaoli county, Taiwan).

### 4.4. Induction of Type 1 Diabetes Using Alloxan

In this step the protocol described in Reference [48] with slight modification was followed. The fish fasted overnight before alloxan injection. The anaesthetized fish were injected intraperitoneally with 1% (*w*/*v*) alloxan in citrate buffer (0.1 M, pH 4.7) at different doses (100, 200, 250, 300 and 350 mg/kg). Healthy control fish were injected with only citrate buffer (0.1 M, pH 4.7). After the injection, the fish was treated with sucrose according to the following procedure: The fish was immersed in 1% sucrose for 6 h, and then transferred to fresh water for 3 h, where the water was refreshed every 60 min to ensure that all sucrose was cleared from the fish. The fish were then fed a normal diet and maintained overnight. 

This procedure was repeated until blood was collected from the last zebrafish group. Blood was collected every 24 h after the sucrose treatment as mentioned above. The fish were fasted for 24 h prior to blood withdrawal. Finally, the blood glucose level was measured using a glucometer.

### 4.5. Treatment of the Fish

The fish were divided into 4 groups containing 10 fish in each group. Three groups were injected with alloxan (300 mg/kg). After 24 h following the alloxan injection, the first group was injected intraperitoneally with glibenclamide at the dose of 2 mg/kg. Glibenclamide was dissolved in 1% dimethylsulfoxide (in water) to obtain a final concentration of 100 µM and sonicated for 15 min. The second group was injected with insulin at the dose of 1 U/kg. The last group (the healthy group) was injected with water. Finally, blood was collected from each fish, and the blood glucose level was measured as described in the previous section [49,50].

### 4.6. Preparation of the Plant Extract and Force-Feeding Procedure 

The plant material was obtained from Cermin Nan Gedang at Sarolangun District, Jambi, Indonesia, and identified by a botanist, Shamsul Khamis. It was deposited at Kulliyyah of Pharmacy, International Islamic University Malaysia, Kuantan, Malaysia, with voucher specimen number PIIUM008-2. The fresh leaves of the plant were cleaned with water and then dried at room temperature (27 ± 1 °C) for 7 days. The dried leaves were ground and stored at −80 °C before further treatment. As much as 3 g of the leaf powder was transferred into a 50 mL beaker, 30 mL of distilled water was added, and the mixture was boiled for 15 min until the volume of the water was reduced to approximately 20 mL. The extract was filtered using Whatman no. 1 filter paper, and the supernatant was collected and stored at −80 °C prior to force-feeding into the zebrafish. 

The optimized type 1 diabetic zebrafish model described above was used to test the efficacy of the plant extract. New groups of healthy, diabetic, and positive control zebrafish (*n* = 10) were developed alongside the plant extract treated zebrafish. The fish were fasted overnight prior to force-feeding. The plant extract, at doses of 1, 2 and 3 g/kg, was force fed to the diabetic zebrafish.

### 4.7. Derivatization Procedure of *P. malayana* Extract

The plant extract was derivatized prior to injection into GC-MS following the protocol described in Reference [51]. The extract of 25 mg was added in a 2 mL centrifuge tube, dissolved in 50 μL of pyridine and sonicated (Elma, S 30H Ultrasonic, South Orange, NJ, USA) for 10 min at 30 °C. Methoxyamine HCl (100 μL, 20 mg/mL in pyridine) was added to the sample solution and vortexed. Followed by incubation for 2 h at 60 °C in an incubator shaker (Innova 4000-M1192, Weender Landstr, Goettingen, Germany). The tube was further incubated for 30 min at 60 °C after the addition of 300 μL of *N*-Methyl-*N*-(trimethylsilyl) trifluoroacetamide (MSTFA). Lastly, the solution was filtered via syringe filter and covered with aluminum foil and left to stand overnight at room temperature (25 ± 0.5 °C) prior to injection into GC-MS. 

### 4.8. GC-MS Analysis of *P. malayana* Extract

GC-MS analysis was accomplished following the protocol described in Reference [51] with slight modification. The derivatized sample (µL) was injected in the splitless mode into the GC-MS system, which consisted of an Agilent 6890 GC-MS and an HP 5973 mass selective detector. The DB-5MS 5% phenyl methyl siloxane column with an inner diameter (ID) of 250 μm and a film thickness of 0.25 μm (Agilent Technologies Inc, Santa Clara, CA, USA) was used. The initial oven temperature was set to 85 °C, and then increased to a target temperature of 315 °C at a rate of 2 °C/min with a total running time of 120 min. Helium was used as the carrier gas with a flow rate of 1 mL/min. The injector and ion source temperatures were set to 250 and 280 °C, respectively. Mass spectra were acquired using a full scan and a monitoring mode with a mass scan range of 50 to 550 *m*/*z*. The spectra for each of the chromatogram peaks were compared with those in the NIST14 database library. The chromatogram and mass spectra were processed using an Agilent ChemStation, Automated Mass Spectral Deconvolution and Identification System and Agilent’s Deconvoluted Reporting Software (Agilent Technologies Inc, Santa Clara, CA, USA). 

### 4.9. LC-MS-Q TOF Based Fingerprinting

The zebrafish blood was centrifuged at 10,000 rpm for 10 min to separate the serum (supernatant) from the debris (precipitate). The serum was then immersed in liquid nitrogen for enzyme inactivation. Subsequently, 5 µL of the serum was added to 250 µL of water: methanol (1:1, *v*/*v*), vortexed and centrifuged under 10,000 rpm for 10 min. Then, 200 µL of the supernatant was transferred into the LC-MS insert vial and stored at −80 °C prior to the LC-MS analysis. Ten microliters of the sample were injected into the LC-MS-Q TOF system. A C_18_ column (Phenomenax Kinetex core-shell technology 100 Å, 250 mm × 4.6 mm, 5µm) was used as the stationary phase. The solvent system was a gradient from 5 to 100% methanol in water with 0.1% phosphoric acid for 10 min and then continued with absolute methanol for another 10 min with a flow rate of 0.3 mL/minute. Electrospray ionization in the positive mode without fragmentation was applied. The voltages of the capillary and sampling cone was set at 3 kV and 40 kV, respectively. The desolvation flow was set to 700 L/h at 300 °C, while the source temperature was set to 110 °C. The MS data were collected in the range of *m*/*z* 100 to 1000 with a scan time of 0.2 s and an interscan delay time of 0.02 s. Leucine-enkephalin (556.2771 Da; 200 mole) in ESI positive mode, was used as the lock spray for the analysis with a flow rate of 3 µL/min [52].

The data was pre-processed using ACD/Spec Manager v.12.00 lab software (Advanced Chemistry Development, Inc., ACD/Labs Toronto, Canada) and converted into cdf file. All spectra were further re-processed for peak filtering, peak identification, peak matching, retention time correction and peak filing using the online MZmine software and converted to an excel format file. Finally, the data was statistically calculated using the SIMCA 14.0 software (Umeå, Sweden).

### 4.10. Histological Study

#### 4.10.1. Organ Harvesting and Fixation

The method which uses 10% formaldehyde as the main fixative agent was applied in this study in accordance with the established protocol [53,54,55] with some modifications. The fish were sacrificed using hypothermal shock through immersion in water at 0 °C for 3 min [56]. Following euthanasia, the fish were gently handled and dried using a tissue paper and immediately preserved in 10% formaldehyde for 24 h. On the following day, the fish were removed from the formaldehyde solution and the whole internal organs except the heart and kidney were harvested and preserved in the formaldehyde solution for another 24 h.

#### 4.10.2. Tissue Processing

The organ’s tissues were then exposed to the consecutive immersion of the tissue through various concentration of ethanol (50%, 70%, 80%, 95%, and 100%, *v*/*v*) for 40–60 min each. Subsequently, the tissues were exposed to ethanol -toluene solution (1:1) for 15 min and followed by the overnight incubation in the absolute toluene. In the following day, the tissues were exposed to three distinctive paraffin waxes for 1 h each under 60 °C in an oven. Upon completion, the tissues were inserted into paraffin wax to facilitate desired orientation for sectioning.

#### 4.10.3. Tissue Sectioning and Hematoxyline and Eosin (H and E) Staining

The tissue was cut using Leica RM2135 microtome which set at 6 µm thickness and subsequently placed on the slides. The tissue slides were then dried on a hot plate at 37–42 °C for 24 h. After that, the slides were consecutively dipped two times in the absolute xylene for 5 min respectively. Subsequently, they were serially placed in different concentrations of ethanol (100%, 90%, 80%, 70% and 50%) for 3 min in each concentration. Following different exposure to different ethanol concentration, the tissue slides stained with H and E as per usual practice in histology i.e., through exposure to haematoxyline for 15 min. The tissue slides were then cleaned under running water for approximately 10 min, followed by immersion in 1% acid alcohol to expel the excess of haematoxylin. The second recoloring step was applied through the exposure of the slides in an eosin solution for 3 s. After that recent exposure, they were again immersed in different concentration of ethanol (95%, 95% 100%, 100%) for 3 min in each concentration, followed by two consecutive immersions in the absolute xylene for 5 min each. Finally, the slide was covered with the layer blend of disteryne, plasticizer and xylene (DPX) mixture before they were cover-slipped. The covered slides were kept under room temperature (27 ± 1 °C) for at least 24 h prior to observation under a light microscope.

## 5. Statistical Analysis

The data were calculated using the one-way analysis of variance (ANOVA) and processed utilizing Minitab version 16 (Coventry, London, UK). Tukey’s test was applied to determine the differences within the groups. The results were presented as mean ± standard deviation, while the results with a p-value below 0.05 were considered significantly different. The multivariate data analysis was performed using SIMCA version 14.0 (Umeå, Sweden). The data was UV scaled and centered prior to fitting. The Partial Least Square-Discriminant Analysis was applied to fit the data. Subsequently, the model was validated through a permutation test. The score scatters plot was used to display the separation among the samples. 

## 6. Conclusions

This study has succeeded in optimizing the induction conditions for the type 1 diabetes zebrafish model. A single injection of alloxan at a dose of 300 mg/kg can induce a type 1diabetes in zebrafish lasts for 7 days after the alloxan injection. Furthermore, insulin injection can be used as a positive control to normalize the serum profile. The reliability of this model was verified by its capability to prove the blood glucose lowering effect of insulin and the *P. malayana* leaf extract in zebrafish. 

The extract was able to lower and normalize the blood glucose levels of diabetic zebrafish. In addition, LC-MS-based fingerprinting indicated that 3 g/kg plant extract shifted the serum metabolite profile of diabetic zebrafish closest to that of the healthy zebrafish compared to the other doses of the plant extract. The mechanism by which this plant lowered the blood glucose level was not through the recovery of the pancreatic cell since this extract could not repair the considerable reduction in pancreatic β-cells. In addition, the plant extract did not alter the healthy liver structure of the zebrafish. This report is the first to describe the anti-diabetic activity of this plant indicating its potential as an anti-diabetic agent. 

## Figures and Tables

**Figure 1 molecules-24-01506-f001:**
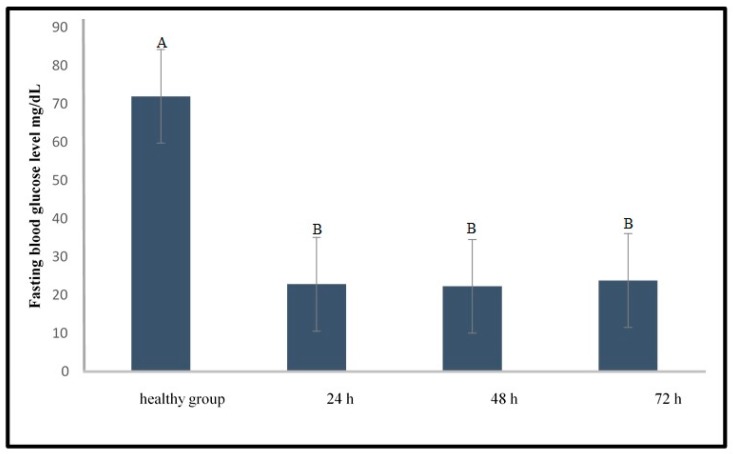
Fasting blood glucose level of healthy and streptozotocin (STZ) treated zebrafish in different timeline following the protocol of Intin et al. [12]. Different capital letter indicates significant difference (*p* < 0.05) among the values, *n* = 10.

**Figure 2 molecules-24-01506-f002:**
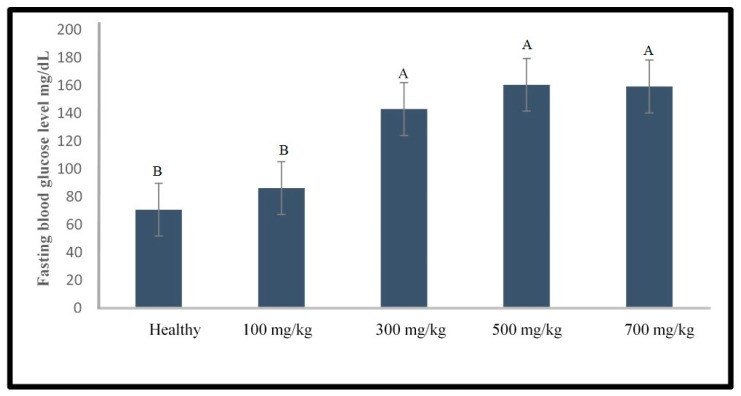
Fasting blood glucose level of zebrafish at 24 h analyzed after a single injection of STZ with various doses. Different capital letter indicates significant difference (*p* < 0.05) among the values, *n* = 10.

**Figure 3 molecules-24-01506-f003:**
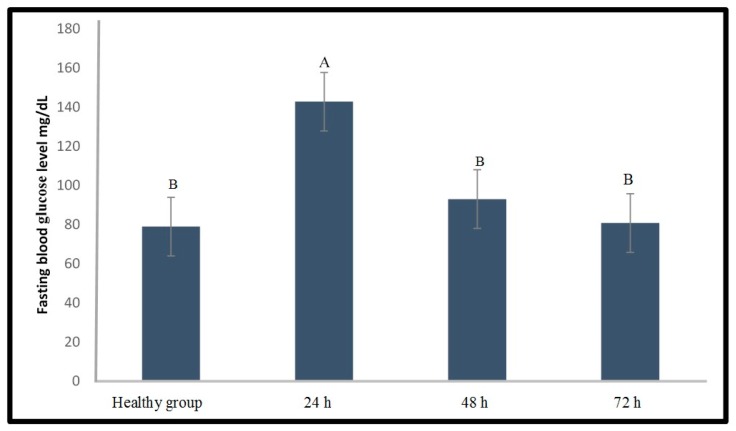
Fasting blood glucose level of zebrafish analyzed at different times after the injection of STZ (300 mg/kg). Different capital letter indicates significant difference (*p* < 0.05) among the values, *n* = 10.

**Figure 4 molecules-24-01506-f004:**
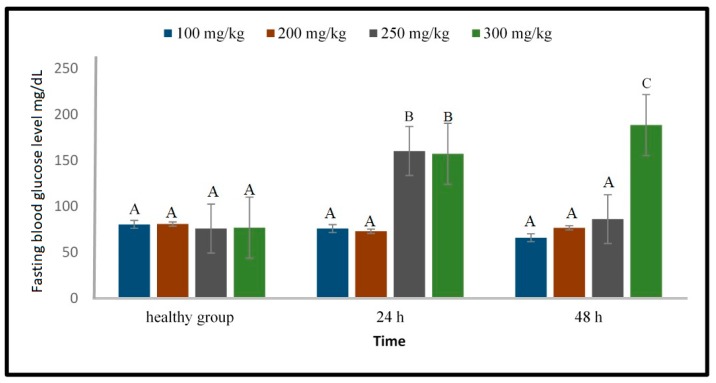
Fasting blood glucose level of zebrafish analyzed after injection of 100, 200, 250 and 300 mg alloxan/kg using 1% sucrose solution. Different capital letter indicates significant difference (*p* < 0.05) among the values, *n* = 10.

**Figure 5 molecules-24-01506-f005:**
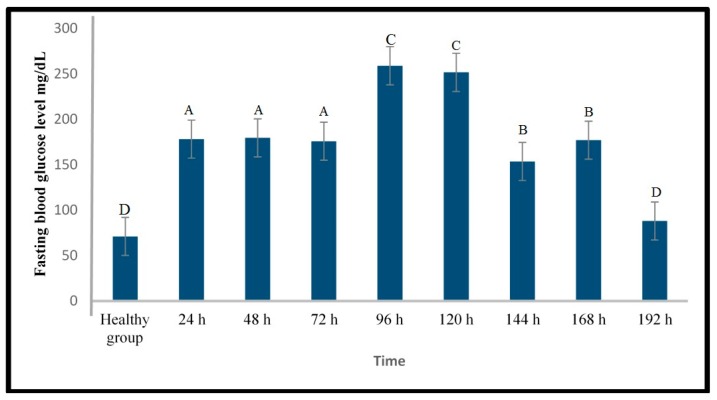
Fasting blood glucose level of zebrafish analyzed after injection of 300 mg alloxan/kg for 8 days. Different capital letter indicates significant difference (*p* < 0.05) among the values, *n* = 10.

**Figure 6 molecules-24-01506-f006:**
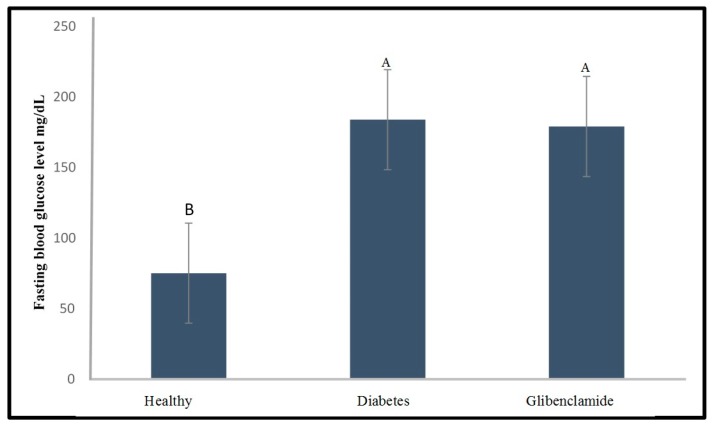
Fasting blood glucose level of the healthy, diabetes, and glibenclamide injected zebrafish (glibenclamide dose = 1.5 mg/kg) at 48 h after the alloxan injection. Different capital letter indicates significant difference (*p* < 0.05) among the values, *n* =10.

**Figure 7 molecules-24-01506-f007:**
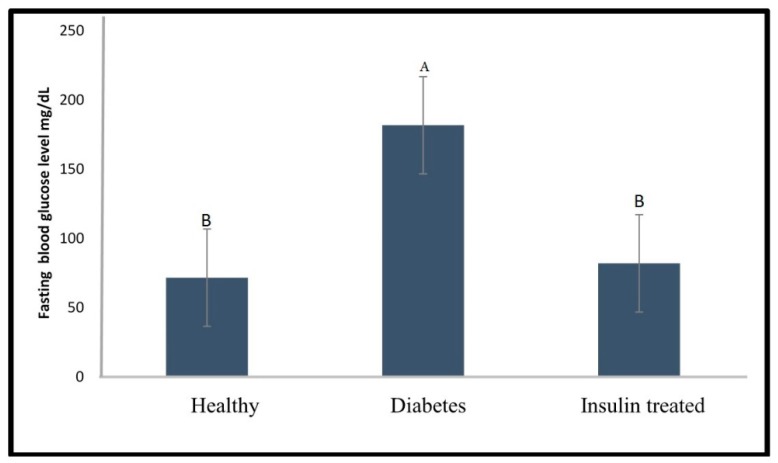
Fasting blood glucose level of healthy, diabetic, and insulin (1 U/kg) treated zebrafish. Different capital letter indicates significant difference (*p* < 0.05) among the values, *n* =10.

**Figure 8 molecules-24-01506-f008:**
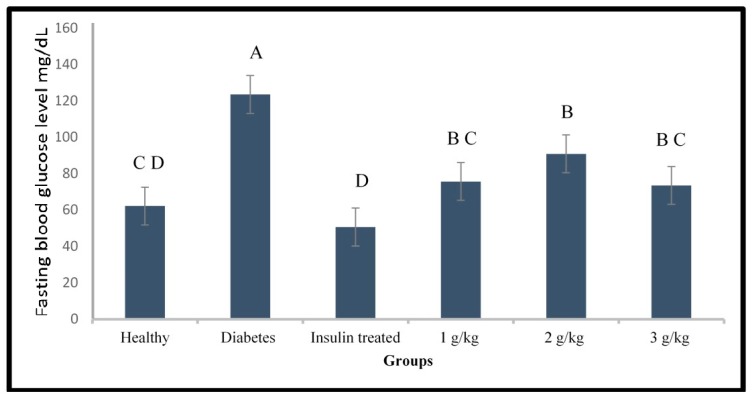
Fasting blood glucose level of diabetic zebrafish after treated with different doses of plant extract, comparing diabetes to healthy and insulin-treated fish. Different capital letter indicates significant difference (*p* < 0.05) among the values, *n* = 10.

**Figure 9 molecules-24-01506-f009:**
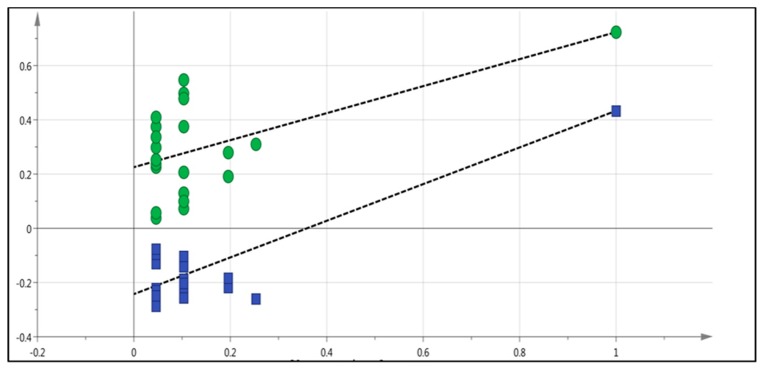
Permutation results of Partial Least Square-Discriminant Analysis (PLS-DA).

**Figure 10 molecules-24-01506-f010:**
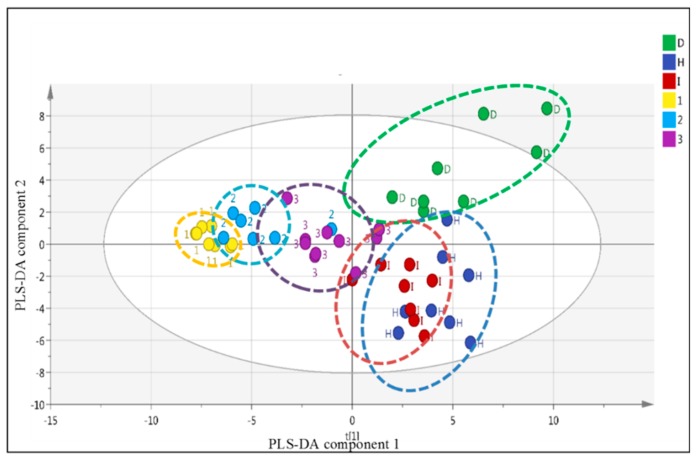
The score scatter plot from the PLS-DA of the healthy group (H), diabetes group (D), insulin-treated group (I), and the plant extract treated groups with the dose of 1 (1), 2 (2), and 3 mg/kg (3) at 24 h after the treatments, and 48 h after the alloxan injection.

**Figure 11 molecules-24-01506-f011:**
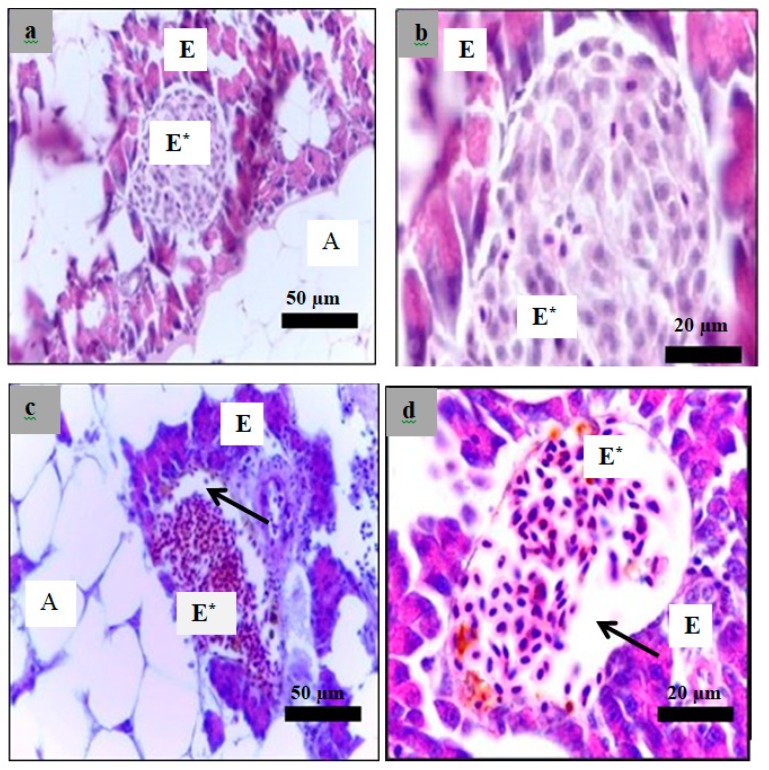
Paraffin section of the islet of the zebrafish pancreas analyzed using hematoxylin and eosin. Healthy group sections (**a**): 40× magnification; (**b**): 100× magnification) showed normal histology of pancreas with high cellularity of Langerhans. Alloxan injected group sections (**c**): 40× magnification; (**d**): 100× magnification) showed a high reduction of endocrine islets cellularity as indicated by the arrows in the alloxan group. E*: endocrine part, E: exocrine part. A: adipose tissue.

**Figure 12 molecules-24-01506-f012:**
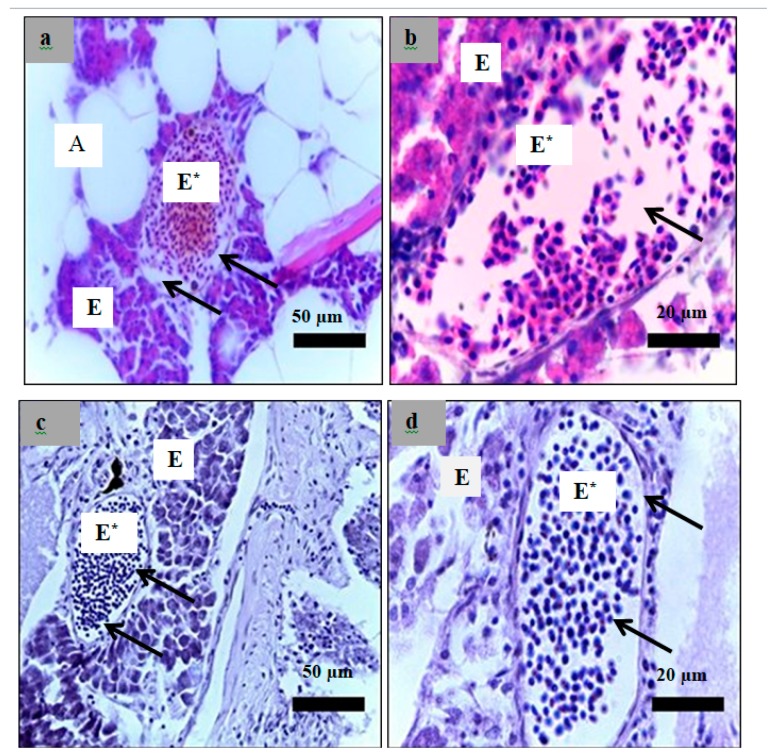
Paraffin section of islet of zebrafish pancreas analyzed using hematoxylin and eosin. (**a**): The insulin-treated zebrafish (40× magnification); (**b**): The insulin-treated zebrafish (100× magnification); (**c**): The plant extract (dose = 3 g/kg) treated zebrafish (40× magnification); (**d**): The plant extract (dose = 3 g/kg) treated zebrafish (100× magnification); The islets showed a high reduction of endocrine islets cellularity as indicated by the arrows. E*: endocrine part, E: exocrine part. A: adipose tissue.

**Figure 13 molecules-24-01506-f013:**
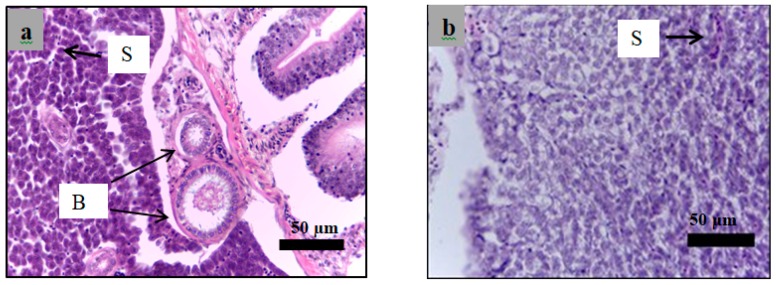
Paraffin section of zebrafish liver analyzed using hematoxylin and eosin staining for the healthy zebrafish ((**a**,**b**) 40× magnification). Section showed the normal hepatocytes structure. B: bile ductules, S: sinusoidal spaces.

**Figure 14 molecules-24-01506-f014:**
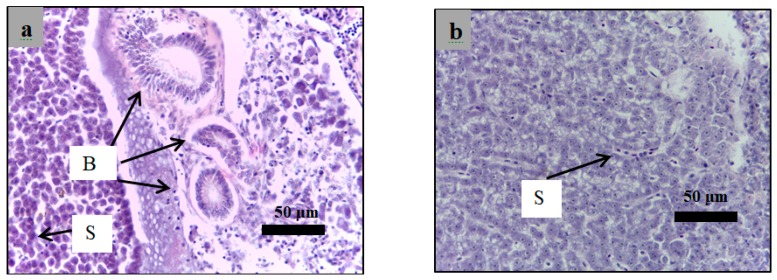
Paraffin section of zebrafish liver analyzed using hematoxylin and eosin staining for the diabetic zebrafish ((**a**,**b**): 40× magnification). Section showed the normal structure of hepatocytes comparing to the control. B: bile ductules, S: sinusoidal spaces.

**Figure 15 molecules-24-01506-f015:**
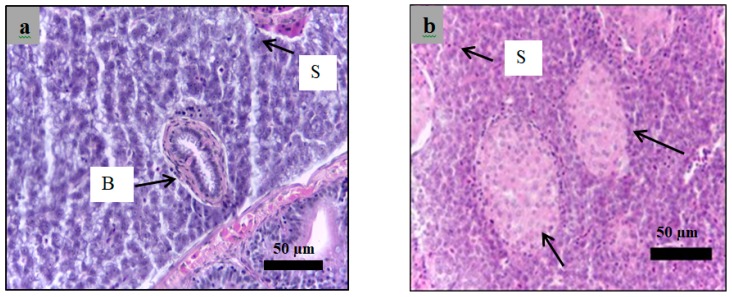
Paraffin section of zebrafish liver analyzed using hematoxylin and eosin staining for the insulin-treated zebrafish ((**a**,**b**): 40× magnification). Nodules of damaged hepatocytes are indicated by arrows. B: bile ductules, S: sinusoidal spaces.

**Figure 16 molecules-24-01506-f016:**
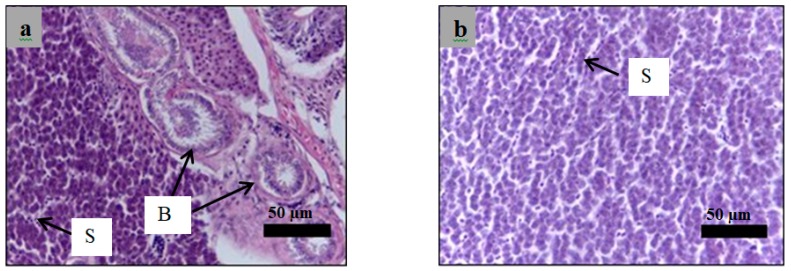
Paraffin section of zebrafish liver analyzed using hematoxylin and eosin staining for the plant extract treated zebrafish ((**a**,**b**): 40× magnification). The hepatocytes show normal structure comparing to control. B: bile ductules, S: sinusoidal spaces.

**Table 1 molecules-24-01506-t001:** Tentative compounds present in the *P. malayana* aqueous leaf extract detected by GC-MS.

Tentative Compounds	Retention Time (min)	Similarity Index	Peak Area (%)
Erythrono-1,4-lactone	21.1	95	0.03
Erythritol	29.4	90	0.09
Xylonic acid	35.9	86	0.02
1,3,5-Benzetriol	37.2	99	0.03
Xylitol	41.6	93	0.09
Shikimic acid	46.7	98	0.49
Quininic acid	48.8	91	1.31
Cyanuric acid	49.0	80	0.02
d-Fructose	49.6	91	4.91
d-Galactose	50.5	91	0.44
d-Allose	50.7	91	0.68
d-Mannose	50.9	91	9.81
Galactose oxime	51.7	93	1.80
l-Sorbose	52.9	93	0.72
3,4-Dihydroxyphenylglycol	56.9	91	0.08
Palmitic acid	57.9	99	0.01
Myo-inositol	60.1	90	5.94
Stearic acid	67.3	98	0.05
Cholesta-7,9(11)-dien-3-ol	78.3	95	1.09
1-Monopalmitin	82.0	93	0.14
Sucrose	84.1	93	33.53
d-Trehalose	87.8	93	0.43
Glycerol monostearate	89.3	90	0.10
Supraene	90.3	93	0.10
4-(1*H*-Pyrrol-1-yl)-piperidine	98.3	93	0.08
Alpha-tocopherol	101.2	98	0.22
Beta-sitosterol	107.2	99	0.27
5-Beta-cholest-24-en-12-one	108.5	90	0.07

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
