# Peer review of "Optimization of Hyperglycemic Induction in Zebrafish and Evaluation of Its Blood Glucose Level and Metabolite Fingerprint Treated with Psychotria malayana Jack Leaf Extract"

_molecules, 2019, doi:10.3390/molecules24081506_

Round 1

Reviewer 1 Report

Manuscript entitled “Optimization of hyperglycemic induction in zebrafish and evaluation of its blood glucose level and metabolite fingerprint treated with Psychotria malayana Jack leaf extracts” established a new type 1 diabetes model with zebrafish. Moreover, the type 1 diabetes zebrafish model is useful for evaluation of beneficial effect of Psychotria malayana Jack leaf extracts. The study includes important insight to develop new anti-diabetic drugs by in vivo evaluation system with zebrafish. The alloxan-induced type 1 diabetes model with zebrafish will contribute to search active fractions that are prepared from natural samples. 1. Zebrafish also has an advantage that can be performed genetic study based on transgenic technique. The authors should be better to write the point in introduction. 2. Green back in Figure 1-8 should be changed. It is difficult to see these figures. 3. The authors should change the sentence “STZ induced type 1 zebrafish” in Figure legend of Figure 1. For example, “STZ treated zebrafish”.

Author Response

We agree with the general thrust of the reviewer’s comments– that there were many opportunities to improve the originally submitted manuscript – and have spent considerable time rewriting our original draft to address their concerns as summarized in the following:

1. Comment from the reviewer:

Zebrafish also has an advantage that can be performed genetic study based on transgenic technique. The authors should be better to write the point in introduction.

Our response:

Correction has been made according to the reviewer’s suggestion. We added the suggested sentence in line 50-52

2. Comment from the reviewer:

Green back in Figure 1-8 should be changed. It is difficult to see these figures

Our response:

Corrections on Fig 1-8 have been made accordingly 

3. Comment from the reviewer:

The authors should change the sentence “STZ induced type 1 zebrafish” in Figure legend of  Figure 1. For example, “STZ treated zebrafish”.

Our response:

Correction has been made accordingly

4. Comment from the reviewer:

Moderate English changes required

Our response:

The English has been revised throughout this manuscript.

Reviewer 2 Report

This manuscript describes the reduction of glucose levels by extracts of a medicinal plant (Psychotria). Zebrafish was chosen as an animal model system because of the easy of known operation and the similarity with the human physiology. The manuscript of this detailed study  is well-written but the authors should check their text carefully for errors, like even the Latin name of their title plant (Psychotria on line 30, abstract). Also, it should be 'aims' on the same line as there are several goals. Line 40 should be corrected to 'anti-diabetic activity'. The authors should also add in the abstract that an analysis of the constituents of this plant was peformed (and they should add the gross classes of compounds found). On line 66/67 the authors should add the type of compounds that were described in literature (which alkaloids?).

The 'H' in the pyrrole name on page 7 should be in italic. Fuhhter to that (in the table), the trifluoromethyl compound is certainly not a natural product. The authors should check this fact (MS data) and comment on it. Tt is probably an error or it is a contaminant that should be addressed by the authors. What is the origin?

Medicinal plants are important for conquering diabetes. In this respect, reviews on this theme should be added tot the list of refs: J. Ethnopharm., 92, 1-21 (2004); Natural Prod. Commun., 10, 187-200 (2015).

The results are sound and should be published after minor adaption.

Author Response

We agree with the general thrust of the reviewer’s comments– that there were many opportunities to improve the originally submitted manuscript – and have spent considerable time rewriting our original draft to address their concerns as summarized in the following:

1. Comment from the reviewer:

The manuscript of this detailed study  is well-written but the authors should check their text carefully for errors, like even the Latin name of their title plant (Psychotria on line 30, abstract). Also, it should be 'aims' on the same line as there are several goals. Line 40 should be corrected to 'anti-diabetic activity'.

Our response:

Correction has been made accordingly as shown in line 27 and 37.

2. Comment from the reviewer:

The authors should also add in the abstract that an analysis of the constituents of this plant was performed (and they should add the gross classes of compounds found).

Our response:

Correction has been made accordingly. The suggested sentence has been imbedded in line 35-37.

3. Comment from the reviewer:

On line 66/67 the authors should add the type of compounds that were described in literature (which alkaloids?).

Our response:

Correction has been made accordingly as shown in line 71-72.

4. Comment from the reviewer:

The 'H' in the pyrrole name on page 7 should be in italic. Fuhhter to that (in the table), the trifluoromethyl compound is certainly not a natural product. The authors should check this fact (MS data) and comment on it. Tt is probably an error or it is a contaminant that should be addressed by the authors. What is the origin?

Our response:

Correction has been made according as shown in Table 1. The correct name of the compound is 4-(1H-Pyrrol-1-yl)-piperidine.

5. Comment from the reviewer:

Medicinal plants are important for conquering diabetes. In this respect, reviews on this theme should be added to the list of refs: J. Ethnopharm., 92, 1-21 (2004); Natural Prod. Commun., 10, 187-200 (2015).

Our response:

Correction has been made accordingly. A new sentence is imbedded in line 358.
